# Laterality in Responses to Acoustic Stimuli in Giant Pandas

**DOI:** 10.3390/ani11030774

**Published:** 2021-03-11

**Authors:** He Liu, Yezhong Tang, Yanxia Ni, Guangzhan Fang

**Affiliations:** 1Beijing Key Laboratory of Captive Wildlife Technology, Beijing Zoo, No 137 Xizhimenwai Street, Beijing 100044, China; liuherstar@yahoo.com (H.L.); niyanxia163@163.com (Y.N.); 2Chengdu Institute of Biology, Chinese Academy of Sciences, No.9 Section 4, Renmin Nan Road, Chengdu 610041, China; tangyz@cib.ac.cn

**Keywords:** auditory lateralization, acoustic stimuli, valence-specific, hemispheric dominance, giant panda (*Ailuropoda melanoleuca*)

## Abstract

**Simple Summary:**

Functional lateralization in the auditory system has been widely studied. Accordingly, behavioral laterality responses affected by acoustic stimuli have been observed in many vertebrate species. In this study, we assessed giant pandas’ behavioral responses to different acoustic stimuli in order to examine cerebral lateralization. We concluded that adult giant pandas showed a left-hemisphere bias in response to positive acoustic stimuli. Furthermore, we found the specific valence of cerebral lateralization for different categories of acoustic stimuli, of which some were relevant to the lateralization while others were not relevant. Our findings support an evolutionary strategy that giant pandas process auditory signals similar to other mammals.

**Abstract:**

Cerebral lateralization is a common feature present in many vertebrates and is often observed in response to various sensory stimuli. Numerous studies have proposed that some vertebrate species have a right hemisphere or left hemisphere dominance in response to specific types of acoustic stimuli. We investigated lateralization of eight giant pandas (*Ailuropoda melanoleuca*) by using a head turning paradigm and twenty-eight acoustic stimuli with different emotional valences which included twenty-four conspecific and four non-conspecific acoustic stimuli (white noise, thunder, and vocalization of a predator). There was no significant difference in auditory laterality in responses to conspecific or non-conspecific sounds. However, the left cerebral hemisphere processed the positive stimuli, whereas neither of the two hemispheres exhibited a preference for processing the negative stimuli. Furthermore, the right hemisphere was faster than the left hemisphere in processing emotional stimuli and conspecific stimuli. These findings demonstrate that giant pandas exhibit lateralization in response to different acoustic stimuli, which provides evidence of hemispheric asymmetry in this species.

## 1. Introduction

Cerebral lateralization was first discovered in humans and was initially considered to be a specialization only in humans, for over a century [1,2,3]. Especially in a mammal, the corpus callosum is closely related to functional asymmetry [1]. In recent years, the existence of brain asymmetry has been proven in many species of invertebrates and vertebrates [4,5,6]. From pond snails (*Lymnaea stagnalis*) and honeybees (*Apis mellifera*) to dolphins (*Tursiops truncatus*) and elephants (*Elephas maximus*), behavioral asymmetry has been evidenced in different species and even traced back to 500 million years ago [3]. Consequently, hemispheric asymmetry could be an evolutionary advantage, since it increases neural processing velocity and also enhances cognitive capacity and efficiency of brain functions [7,8].

Auditory laterality in mammals has been widely reported [9,10,11,12]. Primates display right and left hemisphere asymmetry in response to communication sounds [13,14]. In domestic horses, a clear left hemisphere preference is exhibited when processing familiar neighbor calls [15], and a left hemisphere advantage in processing conspecific vocalizations has been observed in domestic cats and dogs [16,17]. Two main models of the mechanism of laterality have been proposed. One model is the right hemisphere hypothesis, in which the right hemisphere is generally dominant in processing all types of emotions; the other model is the valence specificity hypothesis, which emphasizes left hemisphere dominance when processing positive emotions and right hemisphere dominance when processing negative emotions [18,19]. Furthermore, valence hypotheses have been developed for different patterns including a sex valence specificity hypothesis, an approach–withdrawal hypothesis, and the emotional-type hypothesis [20,21,22,23,24]. Several studies have supported the right hemisphere hypothesis [25,26,27] and additional evidence of the valence hypothesis has been obtained [28,29,30,31,32], while there has also been some evidence that did not support the two hypotheses [22,23,24]. Behavioral and neurophysiological characteristics of cerebral lateralization have been identified using emotional valence, sound function, age, and sex [21,33,34]. However, it is still unclear whether there is a universal model to apply for all species, therefore, we chose this rarely studied species to improve our understanding of the two cerebral hemisphere lateralization.

Giant pandas (*Ailuropoda melanoleuca*) usually live in dense forests surrounded by bamboo; thus, their auditory sense is more sensitive than their visual sense [35]. Although the hearing distance is limited to a small range, they can recognize the vocalizations of predators and conspecifics [35,36]. Do giant pandas have a lateralization preference in the face of different types of sounds? Previous studies have revealed that giant pandas have an advanced capacity for acoustic discrimination, which contributes to reproduction; male giant panda bleats contain information about their hormonal quality, and female giant panda chirps advertise their fertile phase [35,37,38]. Male and female giant pandas can adjust their behavioral responses according to the size-related information broadcasted by bleats; moreover, females can obtain information from the male’s vocal characteristics, and males can determine the optimum timing for copulation by the vocal cues from females [35,37,39]. However, auditory laterality has not been reported in the giant panda which evolved eight million years ago. Given that the giant panda is considered to be a special example of adaptive evolution from the late Miocene [40], do they have some evolutionary auditory lateralization specializations?

In the present study, our first aim was to investigate auditory laterality in response to different acoustic stimuli using the head or ear turning paradigm in giant pandas. We assumed that giant pandas, similar to most mammals, exhibit a hemisphere preference when processing acoustic stimuli. Furthermore, we investigated the existence of left dominance or right dominance in giant pandas. The second aim was to compare lateralized responses with different types of acoustic stimuli. We hypothesized that giant pandas would turn their heads or ears toward the right in response to conspecific stimuli, and to the left in response to non-conspecific stimuli. It is predicted that the left hemisphere processes positive sounds, and the right hemisphere processes negative sounds in giant pandas.

## 2. Materials and Methods 

### 2.1. Study Animals

The giant pandas in this study included five males and three females from the Beijing Zoo (Table 1). They were born in captivity and reared individually. Their daily diets consisted of 75−90% fresh bamboo and 10−25% supplemental feed, including steamed bread, apples, and carrots, which were fed to the pandas twice daily between 08:30–09:30 and 15:30–16:30. Water was supplied at various times throughout the day.

### 2.2. Auditory Stimuli

The sounds were comprised of conspecific and non-conspecific stimuli. All conspecific vocalizations were collected from the Chengdu Research Base of the Giant Panda Breeding, which the giant pandas from the Beijing Zoo, as the subjects, had never been familiar with. Vocalization recordings were performed using a directional microphone (Shure VP89M, Shure Inc., Niles, USA) and a digital recorder (Tascam DR-100MKII, Teac Corp., Tokyo, Japan) at a frequency range of 10–40 kHz. The acoustic equipment and a digital video recorder (SONY FDR–AXP55, Sony Corp., Tokyo, Japan) were placed 2.5 m behind the subjects during recording. All collected vocalizations were randomly selected for editing and filtering by Adobe Audition 3.0 software (San Jose, CA, USA). Six types of conspecific vocalizations were obtained, including positive acoustic stimuli (low growl, bleat, and squawk) and negative acoustic stimuli (bark, roar, and strong bark), in which the discrimination of vocalization valence depends on the spectral and temporal characteristics of sounds, and is determined by behavioral analysis and visual inspection of narrow-band speech maps [41]. Since pseudo-replication may affect the conclusions of statistical analyses, multiple stimulus exemplars are recommended for use in ecological and animal behavior studies [42]. In the present study, four vocalizations were acquired from four different individuals by random selection from our dataset for each of the conspecific vocalizations. The non-conspecific acoustic stimuli were comprised of white noise, thunder, and vocalizations recorded from predators, including a leopard (*Panthera pardus*) and dhole (*Cuon alpines*), which were considered to be negative stimuli. White noise was generated for the mean duration of the average of all sound stimuli by Audition 3.0 and contained rising and falling edges of 10 ms. The vocalization of predators, including leopard and dhole, was downloaded from the website http://sc.chinaz.com/tag_yinxiao/BaoJiaoSheng.html (accessed on 31 December 2020). In total, twenty-eight acoustic stimuli (twenty-four conspecific vocalizations and four non-conspecific sounds, i.e., twelve positive acoustic stimuli and sixteen negative acoustic stimuli) were used in the present study. The sound pressure of each stimulus was adjusted to 68 ± 0.5 dB SPL using a sound pressure meter (Aihua, AWA6291, Hangzhou, China) (20 µPa, C-weighting, and fast response) measured at one meter away from the speaker).

### 2.3. Experimental Procedures

Playback experiments were conducted in the home cage of each subject. The giant pandas were allowed to move freely in their home cage. A laptop computer connected to a loudspeaker (SME-AFS, Saul Mineroff Electronics, Elmont, NY, USA) was used to play sound samples back. The speaker was carefully placed behind the subject and aligned with the body axis of the animal, as shown in Figure 1. As soon as the head of the subject was in line with the speaker, a selected acoustic stimulus was broadcast. Each of the twenty-eight stimuli was chosen randomly and presented to the subject with a 5 s interstimulus interval. Each block for broadcasting a selected stimulus lasted 5 min or until the subject responded to the stimulus with a change in the head or ear orientation. The head or ear orientation and response time were recorded with customer-made software for Windows during acoustic playbacks (Figure 2). In advance, an experimenter was trained for objective judgement through a blind paradigm in which the experimenter was required to determine the occurrence of head turning based on a virtual anteroposterior line in the middle of the head or pinna movements. This experimenter was required to pass a reliability test in order to ensure their capability of precise judgement. Some front grids of the cage were applied as a reference between the panda and experimenter so that the experimenter, grid, and animal were aligned spatially. The responses were scored immediately in the blind protocol and the results were confirmed by researchers on site. The procedure was repeated until all stimuli had been presented; each stimulus was used only once for each subject. If there were some ineffective responses to the continuous and repetitive stimuli, these trials were repeated after several days. All playback orders were randomized with custom software in C++ and saved as .txt files. Test sessions were carried out from April 2018 to June 2018 and did not occur during feeding time between 14:00 and 17:00 each day. Only one subject panda was tested periodically during the day. Whenever a giant panda turned its head to one side (left or right), a lateralized response was recorded. We considered the first response of ear or head orientation towards the loudspeaker to be a valid response and measured the latency of ear or head turning towards the loudspeaker. Finally, the sum of responses for all subjects responding to each stimulus and the latencies for each stimulus were compiled for further statistical analyses.

### 2.4. Statistical Analyses

All data were evaluated for assumptions of normality using the Shapiro–Wilk test and homogeneity of variance using the Levene’s test. As the data failed to meet the statistical assumptions, nonparametric tests were further applied in SPSS 21.0 (SPSS Inc., Chicago, IL, USA). To rule out possible pseudo-replication effects, Fisher’s exact test was used to investigate differences among the four stimulus exemplars for each conspecific vocalization. No significant difference in orientation bias between the left and right was observed for these stimulus exemplars, consistent with the idea that the results of the present statistical analyses were not affected by pseudo-replication. Thus, all datasets were pooled regardless of “stimulus exemplar” and statistically analyzed using a single-sample chi-square test to analyze differences between the numbers of orientations elicited for “positive stimulus”, “negative stimulus”, “conspecific stimulus”, and “non-conspecific stimulus”. The Wilcoxon signed ranks test was chosen to examine the differences in responsive latency among stimuli. The effect size was determined with Cohen’s *w* and *r* for the single-sample chi-square test and Wilcoxon signed ranks test, respectively; small, medium, and large effects were based on value of 0.10, 0.30, and 0.50, respectively [43]. A significance level of *p* < 0.05 was used in all comparisons, and *p* < 0.1 was considered to be marginally significant.

### 2.5. Ethics Approval

This study was approved by the faculty at the Beijing Zoo and the National Bureau of Forestry. This work was carried out following the standard animal care guidelines and did not affect the management practices and decisions made by the zoo.

## 3. Results

### 3.1. Emotional Effect

When the positive emotional stimuli (low growl, bleat, and squawk) were played, the number of right orientation responses was higher than that of the left orientation responses (single-sample chi-square test, χ^2^ = 4.419, *p* = 0.042, Cohen’s *w* = 0.218) (Figure 3). When the negative emotional stimuli (bark, roar, strong bark, white noise, thunder, and vocalization from predators) were played, there was no significant difference between the numbers of left and right orientation responses (single-sample chi-square test, χ^2^ = 0.669, *p* = 0.413, Cohen’s *w* = 0.074). The latency of the response to negative stimuli was significantly shorter than that of the response to positive stimuli (Wilcoxon signed ranks test, *Z* = −2.277, *p* = 0.023, Cohen’s *r* = 0.158) (Figure 4), especially for turning to the left side, and the latency of the response to negative stimuli was significantly shorter than that of the response to positive stimuli (Wilcoxon signed ranks test, *Z* = −2.841, *p* = 0.004, Cohen’s *r* = 0.286) (Figure 4). During the playback of the positive stimuli, the latency of the response of turning to the left response was significantly shorter than that of the turning to the right response (Wilcoxon signed ranks test, *Z* = −2.292, *p* = 0.022, Cohen’s *r* = 0.246) (Figure 4). Similarly, the latency of the turning to the left response was also significantly shorter than that of the turning to the right response during the negative-stimuli playback (Wilcoxon signed ranks test, *Z* = −2.660, *p* = 0.008, Cohen’s *r* = 0.242) (Figure 4).

### 3.2. Conspecific and Non-Conspecific Effects

In response to the playback of conspecific vocalization, although the latency of the response of turning to the right side was significantly longer than that of the response of turning to the left side (Wilcoxon signed ranks test, *Z* = −2.443, *p* = 0.015, Cohen’s *r* = 0.183) (Figure 4), no significant difference was found in the number of right and left orientations towards the loudspeaker (single-sample chi-square test, χ^2^ =1.257, *p* = 0.262, Cohen’s *w* = 0.084) (Figure 3). For the playback of non-conspecific stimuli, there was a marginally significant difference in the latency of the responses of turning to the right and left (Wilcoxon signed ranks test, *Z* = −1.807, *p* = 0.071, Cohen’s *r* = 0.336, marginally significant), but there was no significant difference in the number of orientations towards the right and left (single-sample Chi-square test, χ^2^ = 0.862, *p* = 0.353, Cohen’s *w* = 0.172). Furthermore, no significant difference in the latencies of responses was found between playbacks of conspecific and non-conspecific stimuli (Wilcoxon signed ranks test, *Z* = −0.836, *p* = 0.403, Cohen’s *r* = 0.058). For left bias, the latency of responses to non-conspecific sounds was marginally significantly shorter than that of responses to conspecific sounds (Wilcoxon signed ranks test, *Z* = −1.859, *p* = 0.063, Cohen’s *r* = 0.187, marginally significant) (Figure 4). For right bias, there was no significant difference between the latency of responses to conspecific and non-conspecific sounds (Wilcoxon signed ranks test, *Z* = −1.219, *p* = 0.223, Cohen’s *r* = 0.117).

## 4. Discussion

This study provides evidence of auditory laterality in giant pandas. The giant pandas responded by turning their heads or ears to the right after hearing positive acoustic stimuli, suggesting the left hemisphere processes positive emotions. In contrast, upon hearing negative stimuli, the giant pandas responded by turning their heads or ears to the left, but the trend was not significant. A possible hypothesis for the fact that the data did not reach statistical significance is that the sample size was limited. We realize that the small sample size restricted generalizations at this stage. However, giant pandas are very rare in captivity. Eight individuals provide only limited information, and further research is needed to determine the laterality of giant pandas. 

Previous studies have suggested that hemispheric asymmetry was only involved in the human brain since the left hemisphere is dominant in processing speech and language functions, which do not apply to animals (as discussed by MacNeilage et al. (2009) and Rogers (2010)) [31,44]. However, the left hemisphere has undergone specialization for vocal production and auditory processing in vertebrates and invertebrates [1,4,45]. Regarding the evolution of the hemispheres, brain laterality has been considered to promote fitness benefits for improved cognitive performance [1,46]. Hemisphere preferences in carnivores have been investigated in several families, such as Canidae and Felidae [47]. The majority of studies have reported auditory processing asymmetries focused on cats and dogs [16,17,47]. In Ursidae, there have been several studies on limb preference that have reported that a significant individual-level asymmetry was observed in black bears (*Ursus americanus kermodei*) and the giant panda, but no significant population-level asymmetry was found for the laterality of paw preference [48,49]. From previously reported studies, no auditory lateralization research has been conducted on giant pandas.

The valence hypothesis supposes that vigilance to predators is left biased, and there is a right bias to positive stimuli [50,51]. Although the perception of positive emotions was consistent with the valence hypothesis, our findings did not completely support either the valence hypothesis or the right hemisphere hypothesis. We assumed that there would be a difference in hemispheric functions among various species. Herbivores generally take longer to feed than omnivores and carnivores. Consequently, herbivores have more opportunities to be exposed to danger. Since danger-related negative emotions are considered to be a phylogenetically old emergency response of the brain [52], danger-related emotions may be processed in both of the hemispheres. Our results suggest that there was no significant difference in the processing of negative stimuli between the two hemispheres. When herbivore giant pandas feed on bamboos in the wild, they simultaneously process danger signals with both hemispheres. This can be explained by the fact that evolution has driven the whole brain to respond to an emergency. 

Moreover, the latency of the response to negative stimuli was faster than that of the response to positive stimuli; this may suggest that the brain has evolved to process danger signals quickly to facilitate escape. Since danger is a stronger selective pressure, the response to negative stimuli must be faster. Consequently, the pressure may be an important driver of evolution. It is well known that the right hemisphere plays an essential role in the response to emotional processes [26,53]. Despite a lack of evidence to support the right hemisphere hypothesis, the latency of the response of the right hemisphere was faster than that of the response of the left hemisphere when exposed to emotional stimuli and conspecific stimuli. This is consistent with previous studies that have shown that the right hemisphere processed auditory signals more rapidly than the left hemisphere [54,55]. Although our results did not support that the right hemisphere controls negative stimuli such as the studies on other vertebrates [54,55,56], the negative stimuli may allow an animal to use the higher brain area for escaping potential threats, while the right hemisphere could respond more rapidly to confront these changes [57,58].

Our results revealed no orientation difference between responses to conspecific and non-conspecific stimuli, but the reaction time for responding to non-conspecific stimuli was faster than that for responding to conspecific sounds. This result is consistent with findings in mouse lemurs (*Microcebus myoxinus*) and barbary macaques (*Macaca sylvanus*), which showed no orientation preferences in response to conspecific or heterospecific vocalizations [21,59]. Nevertheless, a considerable number of studies have demonstrated that the left hemisphere is dominant in the processing of conspecific sounds [10,11,14,43,60,61]. Additionally, right hemisphere dominance was observed in response to conspecific vocalization [33,34,62]. The inconsistencies in hemisphere dominance may be explained by differences in stimulus functions, such as familiar versus nonfamiliar, positive versus negative, and social versus individual functions [15,63].

## 5. Conclusions

Our experiments revealed that giant pandas were similar to other mammals regarding processing and responding to acoustic stimuli. We confirmed that giant pandas have auditory laterality when responding to positive stimuli. The results partially support the valence-specific hypothesis and are consistent with proposed evolution strategies. Finally, the results suggest that hemispheric asymmetry and brain laterality might be evolutionarily adaptive for survival and reproduction in the giant panda.

## 6. Limitations

Although there are numerous advantages in this current study, we cannot exclude several limitations. First, the sample size is small, and therefore it is difficult to find significant relationships from the data. Secondly, we did not measure the response scoring using offline video. However, this limitation should not affect the conclusions, since the response scoring was performed on site. Finally, it should be pointed out that the number of acoustic stimuli was unbalanced for the different categories. The limitations of the study can serve as an important opportunity to describe the need for further research.

## Figures and Tables

**Figure 1 animals-11-00774-f001:**
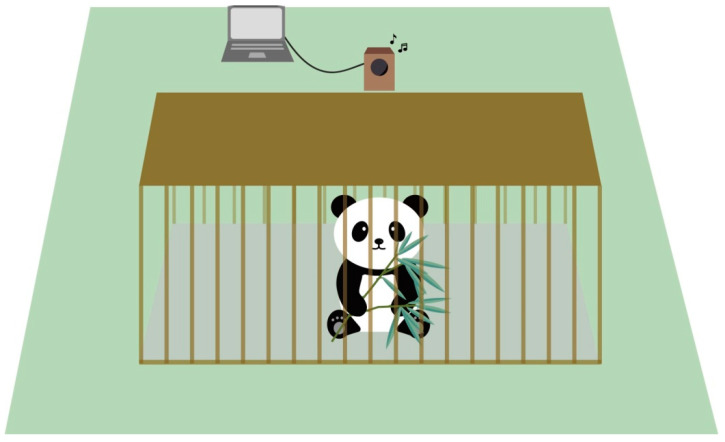
Schematic illustration of the playback setup. Playback experiments were conducted in the home cage of each giant panda where they were free to move. The acoustic equipment and the digital video were placed behind the subjects during recording. As soon as the head of the subject was in line with the speaker, a selected acoustic stimulus was played. The response was recorded when the subjects turned their ears or head towards the speaker.

**Figure 2 animals-11-00774-f002:**
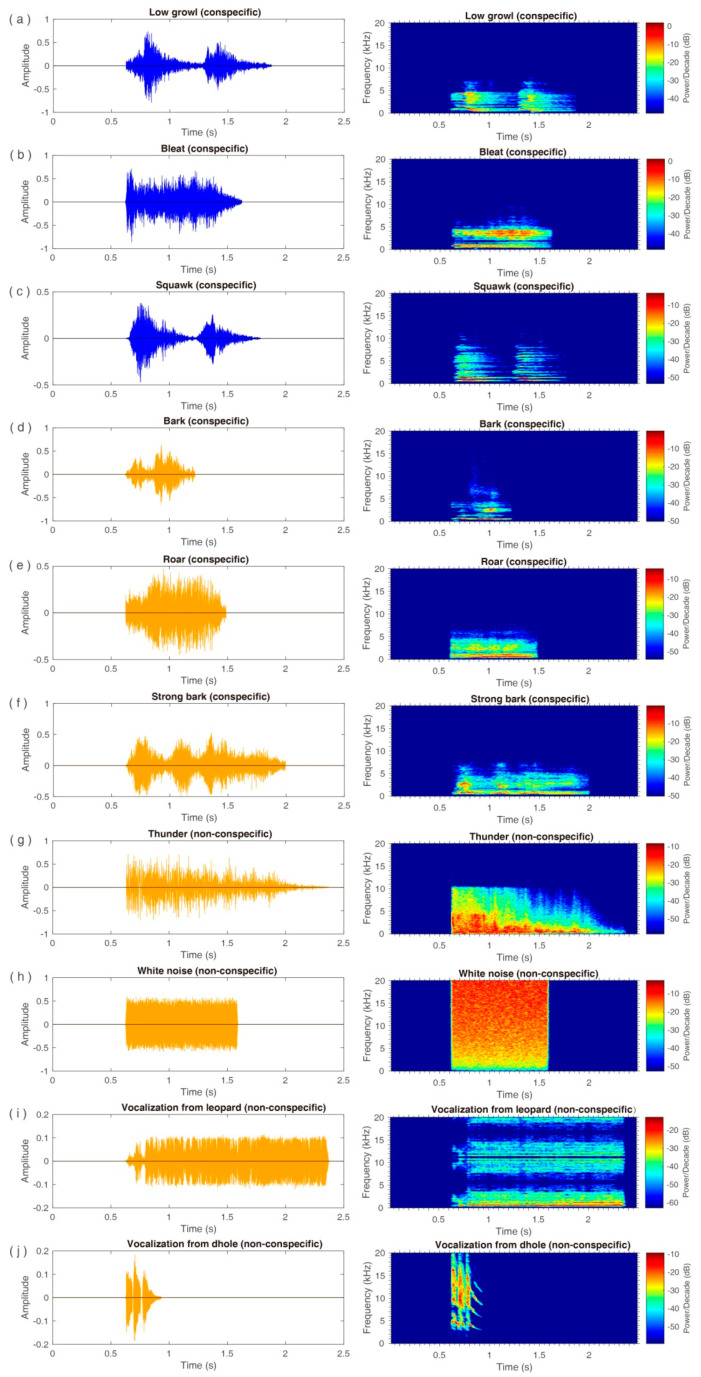
Waveforms and spectrograms of the ten types of stimuli. (**a**) Low growl; (**b**) Bleat; (**c**) Squawk; (**d**) Bark; (**e**) Roar; (**f**) Strong bark; (**g**) Thunder; (**h**) White noise; (**i**) Vocalization from leopard; (**j**) Vocalization from dhole. The blue waves represent the positive stimuli, while the orange waves represent the negative stimuli. Note that only one exemplar is shown for conspecific vocalization, while four exemplars were used for each conspecific sound.

**Figure 3 animals-11-00774-f003:**
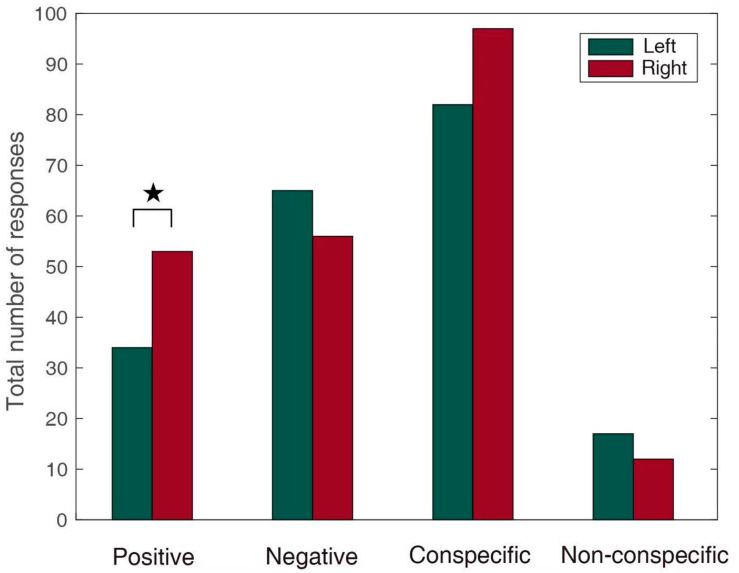
The total number of head or ear orientations of giant pandas in response to different stimuli. The green bar represents a left turn and the red bars represents a right turn. The numbers of right head and ear orientation responses were higher than those of left orientation responses for the positive stimuli (single-sample chi-square test, ★ *p* < 0.05).

**Figure 4 animals-11-00774-f004:**
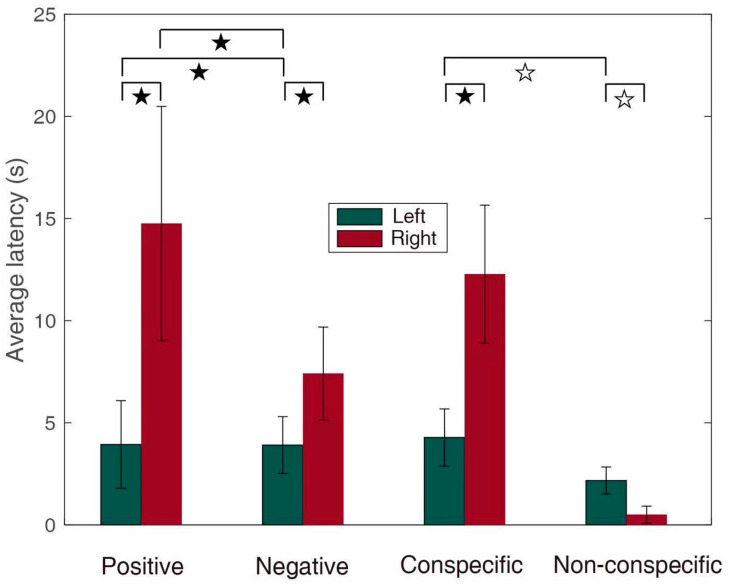
The latencies of head and ear orientation responses to different stimuli in giant pandas. The bars represent average response latencies (mean ± SEM). The stars represent the significance of latencies for the different stimuli (★ *p* < 0.05, ☆ *p* < 0.1).

**Table 1 animals-11-00774-t001:** Information on giant pandas at the Beijing Zoo.

Name	Sex	Age
DADI	♂	Adult
JINI	♀	Adult
GUGU	♂	Adult
FULU	♀	Juvenile
MENGDA	♂	Juvenile
MENGER	♂	Juvenile
DIANDIAN	♀	Juvenile
MENGLAN	♂	Juvenile

## Data Availability

The datasets used and/or analyzed in current study are available from the corresponding author on reasonable request.

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
