# Peer review of "Laterality in Responses to Acoustic Stimuli in Giant Pandas"

_animals, 2021, doi:10.3390/ani11030774_

Round 1

Reviewer 1 Report

Review of “Laterality in Responses to Acoustic Stimuli in Giant Pandas”

In their article “Laterality in Responses to Acoustic Stimuli in Giant Pandas” investigate auditory laterality in Giant Pandas. This is an interesting study in line with the thematic scope (although it would also fit very well with the Journal Symmetry).

General:

  • The manuscript should be proofread by a native English speaker. There are too many grammatical mistakes in it for an international peer reviewed journal

Introduction:

  • Statement: “Cerebral lateralization was first discovered in humans and was initially considered a specialization in only humans for over a century.” The literature cited in the context of comparative research on hemispheric asymmetries should be updated to include the most recent review articles and books, e.g.: Güntürkün O, et al. Brain Lateralization: A Comparative Perspective. Physiol Rev. 2020 Jul 1;100(3):1019-1063. doi: 10.1152/physrev.00006.2019 and Ocklenburg S, Güntürkün O. (2018). The Lateralized Brain. The Neuroscience and Evolution of Hemispheric Asymmetries. Academic Press.
  • Statement “In recent years, the existence of brain asymmetry has been proven in many species of invertebrates and vertebrates” – please included systematic analyses on this topic, e.g. Ströckens F, et al.. Limb preferences in non-human vertebrates. Laterality. 2013;18(5):536-75. doi: 10.1080/1357650X.2012.723008 and Ocklenburg S, et al.. Lateralisation of conspecific vocalisation in non-human vertebrates. 2013;18(1):1-31. doi: 10.1080/1357650X.2011.626561
  • Ailuropoda melanoleuca: Species names should be written in Italics
  • Please include clear hypotheses derived from the literature at the end of the introduction

Methods:

  • Only 8 animals were tested. This should be discussed as a weakness of the study in the discussion section in greater detail.

Results:

  • Please include error bars in Figure 3

Discussion:

  • It would be great if the authors could include a more detailed discussion of the relevance of their findings in the context of previous findings on the lateralization in Ursidae and the Carnivora order in general

Author Response

Point 1: General: The manuscript should be proofread by a native English speaker. There are too many grammatical mistakes in it for an international peer reviewed journal.

Response 1: Thanks for your suggestion. We have checked English carefully.

Point 2: Introduction: Statement: “Cerebral lateralization was first discovered in humans and was initially considered a specialization in only humans for over a century.” The literature cited in the context of comparative research on hemispheric asymmetries should be updated to include the most recent review articles and books, e.g.: Güntürkün O, et al. Brain Lateralization: A Comparative Perspective. Physiol Rev. 2020 Jul 1;100(3):1019-1063. doi: 10.1152/physrev.00006.2019 and Ocklenburg S, Güntürkün O. (2018). The Lateralized Brain. The Neuroscience and Evolution of Hemispheric Asymmetries. Academic Press.

Response 2: Thanks for your kind suggestion. We have read this article about brain lateralization and referred this review to our manuscript in line 35-41.

Point 3: Statement “In recent years, the existence of brain asymmetry has been proven in many species of invertebrates and vertebrates” – please included systematic analyses on this topic, e.g. Ströckens F, et al.. Limb preferences in non-human vertebrates. Laterality. 2013;18(5):536-75. doi: 10.1080/1357650X.2012.723008 and Ocklenburg S, et al.. Lateralisation of conspecific vocalisation in non-human vertebrates. 2013;18(1):1-31. doi: 10.1080/1357650X.2011.626561

Response 3: Thanks for your careful consideration. We have added more details to analyze the systematic asymmetry in line 38-41.

Point 4: Ailuropoda melanoleuca: Species names should be written in Italics Response 4: Thanks for your correction. We are very sorry for this mistake. We have corrected all Latin names to italics.

Point 5: Please include clear hypotheses derived from the literature at the end of the introduction

Response 5: Thanks for your comments. We have clarified the hypotheses from these references in line 79-83.

Point 6: Methods: Only 8 animals were tested. This should be discussed as a weakness of the study in the discussion section in greater detail. Response 6: Thank you, we have discussed this small sample size in the discussion of line 215-218.

Point 7: Results: Please include error bars in Figure 3

Response 7: Thank you for your suggestion, we are very sorry that we filled the wrong vertical ordinate to “Mean number of responses”. Actually it is total number of responses for all pandas, so there are no error bars in this figure. We have corrected the Figure in this version.

Point 8: Discussion: It would be great if the authors could include a more detailed discussion of the relevance of their findings in the context of previous findings on the lateralization in Ursidae and the Carnivora order in general.

Response 8: Thanks for your comment. We referred the findings of Ursidae to the discussion of line 224-231.

END

Reviewer 2 Report

Laterality in Responses to Acoustic Stimuli in Giant 3 Pandas

Peer review:

The paper is interesting and I recommend to publish it, if the necessary revisions would be made:

-The Latin name is not in italics as it should be.

-First citation is wrong. Cerebral lateralization was first found in “lower” vertebrates, not in humans. See in Vallortigara, G. (2000). Comparative neuropsychology of the dual brain: A stroll through animals’ left and right perceptual worlds. Brain and Language, 73, 189–219.

See also: Sion, G., Tal, R., & Meiri, S. (2020). Asymmetric Behavior in Ptyodactylus guttatus: Can a Digit Ratio Reflect Brain Laterality?. Symmetry12(9), 1490.‏

‘’ The focus of a majority of research by neurobiologists in brain asymmetries and brain laterality has mainly been with humans and other primates. This focus is somewhat surprising given that brain laterality was first described in fish, amphibians and reptiles’’

Also:

- Brain lateralization was demonstrated in the so-called lower vertebrates as fish, amphibians, and reptiles, revealing anatomical asymmetries in the diencephalic regions (review in Harris, Guglielmotti, & Bentivoglio, 1996; Vallortigara, 2000). These early data had been, however, largely neglected for many years (see Bisazza, Rogers, & Vallortigara, 1998 for a review).

A lack of important literature:

-Rogers and Anson (1979) found that the right hemisphere (left ear, left eye, etc…) in many vertebrates, is specialized in vigilance toward predators. While left hemisphere (right ear, right eye, etc…), is specialized in search for food. Rogers, L.J.; Anson, J.M. Lateralization of function in the chicken forebrain. Pharmacol. Biochem. Behav. 1979, 10, 679–686. Also, discussed in Sion et al. 2020, that cites relevant literature on the matter.

This hemispheric specialization can explain some of the findings and is most relevant to the discussion, but is missing in the MS.

P2 line 55, again, Latin name is not in italics.

P2 line 65, erase ‘’also’’, it is redundant.

P2 line 95+96, again, Latin name is not in italics. So, do it all over the MS, whenever there is a Latin name, write it in italics... As a scientific name should be written…

P3 line 118: Why only one subject panda was tested periodically during the day? Was the result different? It is better to test all pandas both on day and night and later compare statistically.

P6 results, all statistical parameters should be in italics (such as P value and Z).

The results are consistent with the literature (Rogers and Anson, 1979; Sion et al. 2020) that vigilance to predators is left-biased, and a right bias to positive stimuli. The literature should be cited and discussed. However, there is a nice touch to the result of a neutral noise with a neutral result of no bias. Very interesting and consistent with the literature.

Also, since predators are selective drivers of evolution, it is well understood (logical) that the response to negative stimuli (predator) is faster. Since the selective pressure is stronger.

All these should affect the discussion and conclusions.

Good luck.

Author Response

Response to Reviewer 2 Comments

Point 1:  -The Latin name is not in italics as it should be.

Response 1: Thanks for your correction. We are very sorry for this mistake. We have corrected all Latin names to italics.

Point 2: -First citation is wrong. Cerebral lateralization was first found in “lower” vertebrates, not in humans. See in Vallortigara, G. (2000). Comparative neuropsychology of the dual brain: A stroll through animals’ left and right perceptual worlds. Brain and Language, 73, 189–219. See also: Sion, G., Tal, R., & Meiri, S. (2020). Asymmetric Behavior in Ptyodactylus guttatus: Can a Digit Ratio Reflect Brain Laterality?. Symmetry, 12(9), 1490.‏

‘’ The focus of a majority of research by neurobiologists in brain asymmetries and brain laterality has mainly been with humans and other primates. This focus is somewhat surprising given that brain laterality was first described in fish, amphibians and reptiles’’

Also:- Brain lateralization was demonstrated in the so-called lower vertebrates as fish, amphibians, and reptiles, revealing anatomical asymmetries in the diencephalic regions (review in Harris, Guglielmotti, & Bentivoglio, 1996; Vallortigara, 2000). These early data had been, however, largely neglected for many years (see Bisazza, Rogers, & Vallortigara, 1998 for a review).

Response 2: Thank you for your suggestion, we carefully read all relevant literatures, we think the first described specie was humans by tracing to 16th century (see Güntürkün et al. 2020). Güntürkün O, et al. Brain Lateralization: A Comparative Perspective. Physiol Rev. 2020 Jul 1;100(3):1019-1063. doi: 10.1152/physrev.00006.2019

Point 3: A lack of important literature:

-Rogers and Anson (1979) found that the right hemisphere (left ear, left eye, etc…) in many vertebrates, is specialized in vigilance toward predators. While left hemisphere (right ear, right eye, etc…), is specialized in search for food. Rogers, L.J.; Anson, J.M. Lateralization of function in the chicken forebrain. Pharmacol. Biochem. Behav. 1979, 10, 679–686. Also, discussed in Sion et al. 2020, that cites relevant literature on the matter.

This hemispheric specialization can explain some of the findings and is most relevant to the discussion, but is missing in the MS.

Response 3: Thank you for suggestion, we updated the relevant literature in line 232-233.

Point 4: P2 line 55, again, Latin name is not in italics.

Response 4: Thanks for your correction. We are very sorry for this mistake. We have corrected all latin names to italics.

Point 5: P2 line 65, erase ‘’also’’, it is redundant.

Response 5: Thank you for this, we have deleted the also in the revision.

Point 6: P2 line 95+96, again, Latin name is not in italics. So, do it all over the MS, whenever there is a Latin name, write it in italics... As a scientific name should be written…

Response 6: we are sorry for Latin name, we have corrected to be italics.

Point 7: P3 line 118: Why only one subject panda was tested periodically during the day? Was the result different? It is better to test all pandas both on day and night and later compare statistically.

Response 7: Thank you for your questions, yes, it is better to test all pandas in one day. However, we tested 28 typies of sounds for a panda and at least 5 mins for one sound playing, so each panda need at least 140 mins to play all stimuli. Therefore, we only test one panda for one day during 14:00-17:00.

Point 8: P6 results, all statistical parameters should be in italics (such as P value and Z).

Response 8: Thank you for your correction, we have corrected these parameters to be in italics.

Point 9: The results are consistent with the literature (Rogers and Anson, 1979; Sion et al. 2020) that vigilance to predators is left-biased, and a right bias to positive stimuli. The literature should be cited and discussed. However, there is a nice touch to the result of a neutral noise with a neutral result of no bias. Very interesting and consistent with the literature.

Also, since predators are selective drivers of evolution, it is well understood (logical) that the response to negative stimuli (predator) is faster. Since the selective pressure is stronger. 

All these should affect the discussion and conclusions.

Good luck.

Response 9:  Thank you for your kind suggestion, we appreciated your approvals for MS. We cited these classical articles in line 232-233 and added these suggestions in line246-247, 252-256.

END

Reviewer 3 Report

I found this to be a scientifically  

I found this to be a scientifically sound and interesting work investigating auditory lateralization in captive giant pandas. We authors used a well-established experimental paradigm to test hemispheric specializations in processing different types of acoustic stimuli. In general, I made only minor comments throughout the manuscript to improve the clarity. My only significant concern is that in the Discussion authors make a comparison between humans and pandas which contradicts the current understanding of hemispheric functioning. The studies on other mammals should be considered and the paragraph re-written. In the present form, I can’t recommend this part of the manuscript to be accepted for publication. In addition, the sample size is small and this should be mentioned in the abstract. Otherwise, the conclusions may look too general.

Specific comments

L 14-16: These two sentences seem to contradict each other. How can lateralization depend on valence if the left hemisphere is dominant for processing ALL acoustic stimuli? It is at the very beginning of the manuscript and gives a reader a very confusing impression about you results. I strongly recommend you rewrite these sentences.

L 21: It looks weird that you give a number of acoustic stimuli used in the study but doesn’t mention the sample size. The latter is definitely more important than the former.

L 23-25: Your abstract is too brief to give a clear understanding of the study design. For example, it would be important for a reader to know which non-conspecific sounds have been used. You write that lateralization is evident when processing both conspecific and non-conspecific sounds. At the same time, lateralization is valence-dependent. It would be good to give a reader some idea about how your assessed valence in case of non-conspecific sounds.

L 36-36: I can’t find any reason to mention the corpus callosum here. It has no relation to other information in the Introduction. Thus, it is simply useless.

L 44-49: These hypotheses do not specifically consider auditory laterality but describe the division of hemispheric functions in general. See for example Najt, P., Bayer, U. & Hausmann, M. (2013). Models of hemispheric specialization in facial emotion perception—a reevaluation. Emotion, 13, 159–167; or Leliveld, L. M., Langbein, J., & Puppe, B. (2013). The emergence of emotional lateralization: evidence in non-human vertebrates and implications for farm animals. Applied Animal Behaviour Science, 145(1-2), 1–14.

L 50-54: Please have a look at this paper - Fourie, B., Berezina, E., Giljov, A., & Karenina, K. (2020). Visual lateralization in artiodactyls: A brief summary of research and new evidence on saiga antelope. Laterality, 1-24. It summarizes the current knowledge about valence-dependent visual lateralization. It is clear that valence hypothesis is not clearly applicable to the existing empirical evidence.

L 89-91: You provide a reference but it would be even better if you give a brief explanation of how the vocalization valence was estimated in this previous study.

L 200: A word ‘also’ is probably not necessary here as it is a bit confusing.

L 201-202: Potentially, a weak bias in response to negative stimuli may be associated with very limited negative experience in the individuals kept in a zoo. I suppose they did not get a chance to engage in negative social interactions or experience some non-social negative situations (e.g. hear a thunder). Is it a possibility?

218-223: It is a very limited view on the problem. Why is the comparison made only between humans and pandas? There are numerous species of mammals and other vertebrates showing lateralized processing of negative and emergency stimuli. The idea that pandas being wild animals benefit from non-lateralized processing of negative sounds contradicts all the knowledge about hemispheric lateralization. It is known that lateralization facilitate the speed and intensity of the response. It is assumed that lateralized processing of a stimulus allows to faster responding because there is no conflict between two hemispheres acting simultaneously.  See e.g., Lippolis, G., Westman, W., McAllan, B., & Rogers, L. (2005). Lateralisation of escape responses in the stripe-faced dunnart, Sminthopsis macroura (Dasyuridae: Marsupialia). Laterality, 10(5), 457-470; Bonati, B., Csermely, D., López, P., & Martín, J. (2010). Lateralization in the escape behaviour of the common wall lizard (Podarcis muralis). Behavioural brain research, 207(1), 1-6; Austin, N. P., & Rogers, L. J. (2012). Limb preferences and lateralization of aggression, reactivity and vigilance in feral horses, Equus caballus. Animal Behaviour, 83(1), 239-247.

Author Response

Response to Reviewer 3 Comments

Point 1:  Comments and Suggestions for Authors

I found this to be a scientifically sound and interesting work investigating auditory lateralization in captive giant pandas. We authors used a well-established experimental paradigm to test hemispheric specializations in processing different types of acoustic stimuli. In general, I made only minor comments throughout the manuscript to improve the clarity. My only significant concern is that in the Discussion authors make a comparison between humans and pandas which contradicts the current understanding of hemispheric functioning. The studies on other mammals should be considered and the paragraph re-written. In the present form, I can’t recommend this part of the manuscript to be accepted for publication. In addition, the sample size is small and this should be mentioned in the abstract. Otherwise, the conclusions may look too general.

Response 1:  We appreciated your kind suggestion. In this revision, we compared the findings of Ursidae to the discussion of line 224-231. Also, we mentioned the small sample size in the abstract line 21 and discussion line 215-218.

Point 2: Specific comments

L 14-16: These two sentences seem to contradict each other. How can lateralization depend on valence if the left hemisphere is dominant for processing ALL acoustic stimuli? It is at the very beginning of the manuscript and gives a reader a very confusing impression about you results. I strongly recommend you rewrite these sentences.

Response 2: Thanks for your question. We have re-writed the sentence in line 15-16 as your suggestion.

Point 3: L 21: It looks weird that you give a number of acoustic stimuli used in the study but doesn’t mention the sample size. The latter is definitely more important than the former.

Response 3: Thanks for your comment. we have added the sample size in this sentence of line 21.

Point 4: L 23-25: Your abstract is too brief to give a clear understanding of the study design. For example, it would be important for a reader to know which non-conspecific sounds have been used. You write that lateralization is evident when processing both conspecific and non-conspecific sounds. At the same time, lateralization is valence-dependent. It would be good to give a reader some idea about how your assessed valence in case of non-conspecific sounds.

Response 4: Thanks for your good suggestion, we have re-writed the non-conspecific sounds in abstract of line 21-24.

Point 5: L 36-36: I can’t find any reason to mention the corpus callosum here. It has no relation to other information in the Introduction. Thus, it is simply useless.

Response 5: Thanks for your suggestion, we changed this content in line 36.

Point 6: L 44-49: These hypotheses do not specifically consider auditory laterality but describe the division of hemispheric functions in general. See for example Najt, P., Bayer, U. & Hausmann, M. (2013). Models of hemispheric specialization in facial emotion perception—a reevaluation. Emotion, 13, 159–167; or Leliveld, L. M., Langbein, J., & Puppe, B. (2013). The emergence of emotional lateralization: evidence in non-human vertebrates and implications for farm animals. Applied Animal Behaviour Science, 145(1-2), 1–14.

Response 6: Thank you for your suggestion, we have read the two articles and added the reference in line 51-60.

Point 7: L 50-54: Please have a look at this paper - Fourie, B., Berezina, E., Giljov, A., & Karenina, K. (2020). Visual lateralization in artiodactyls: A brief summary of research and new evidence on saiga antelope. Laterality, 1-24. It summarizes the current knowledge about valence-dependent visual lateralization. It is clear that valence hypothesis is not clearly applicable to the existing empirical evidence.

Response 7: Thank you for your suggestion of articles. Yes, the valence hypothesis only applies for some evidences. We added the context as your suggestion in line 51-60.

Point 8: L 89-91: You provide a reference but it would be even better if you give a brief explanation of how the vocalization valence was estimated in this previous study.

Response 8: Thank you for your suggestion, we have given a brief explanation on the classification of vocalization from this previous study in line 102-104.

Point 9: L 200: A word ‘also’ is probably not necessary here as it is a bit confusing.

Response 9: Thank you for your suggestion, we have deleted the also in sentence.

Point 10: L 201-202: Potentially, a weak bias in response to negative stimuli may be associated with very limited negative experience in the individuals kept in a zoo. I suppose they did not get a chance to engage in negative social interactions or experience some non-social negative situations (e.g. hear a thunder). Is it a possibility?

Response 10: Thank you for your comments. Yes, it is, these zoo animals have a less chance to touch with the negative situations.

Point 11: 218-223: It is a very limited view on the problem. Why is the comparison made only between humans and pandas? There are numerous species of mammals and other vertebrates showing lateralized processing of negative and emergency stimuli. The idea that pandas being wild animals benefit from non-lateralized processing of negative sounds contradicts all the knowledge about hemispheric lateralization. It is known that lateralization facilitate the speed and intensity of the response. It is assumed that lateralized processing of a stimulus allows to faster responding because there is no conflict between two hemispheres acting simultaneously.  See e.g., Lippolis, G., Westman, W., McAllan, B., & Rogers, L. (2005). Lateralisation of escape responses in the stripe-faced dunnart, Sminthopsis macroura (Dasyuridae: Marsupialia). Laterality, 10(5), 457-470; Bonati, B., Csermely, D., López, P., & Martín, J. (2010). Lateralization in the escape behaviour of the common wall lizard (Podarcis muralis). Behavioural brain research, 207(1), 1-6; Austin, N. P., & Rogers, L. J. (2012). Limb preferences and lateralization of aggression, reactivity and vigilance in feral horses, Equus caballus. Animal Behaviour, 83(1), 239-247.

Response 11: Thank you for your good suggestion, we have read these papers and cited these opinions. We also deleted the comparison with human and improved this part of discussion in line 224-231, and we also referred those articles as 55-57 in line 253-256.

END

Reviewer 4 Report

The manuscript by Liu et al. “Laterality in Responses to Acoustic Stimuli in Giant Pandas” aims at investigating lateralization for acoustic stimuli in the giant panda. Eight animals were presented with 10 types of stimuli - 6 of conspecifics (3 positive and 3 negative) and 4 of different nature (white-noise and thunder, and 2 negative non-conspecifics vocalizations) - while aligned with the playback speaker in their home-cage. Authors report no differences as for conspecific vs heterospecific sounds but faster left head oriented responses to emotional and conspecifics’ sounds. Despite the interesting issues investigated, I found several criticisms detailed here below following the order in which both major and minor concerns appear in the manuscript.

Introduction

  • Lines 48-9: if the mentioned hypotheses are relevant – and indeed they seem so because authors refer to them also in the discussion – they should be, at least briefly, described.
  • Lines 51-4: it is unclear to me how the two sentences are related, please clarify.

Procedure

Subjects

  • Where were the pandas from? Details are reported for the place in which panda’s vocalizations have been recorded, but no details are available for the pandas that have been tested. If the facility where the test occurred is the same from which recordings were taken, then the playback experiment could have suffered from the familiarity of the vocalizations to the tested pandas (see also a comment about familiarity in the discussion section). They could be uninterested in the sounds if familiarity is an uncontrolled factor at play.

Set-up and dependent variable

  • I wonder why the authors chose to present the stimuli using a single speaker and waiting for the animal to align with it. The standard playback procedure used with other species implies that the animal reaches a fixed position in a standardized way and then two side-speakers broadcast at the same time the same stimulus. In this way, the sound is homogeneous and the lateral response of the animal reveals the differential elaboration carried out by the two hemispheres. The strategy adopted by Liu et al., instead, suffers from unpredictable availability of the animal (how much did the experimenter had to wait before the animal was in the correct position?), errors in alignment (how was determined that the panda was perfectly aligned to the speaker?) and bias in the subsequent response (if already in a biased position, a biased response could be the result of the procedure and not of the hemispheric differential elaboration of the stimuli).
  • The dependent variable was the head or the ear orientation while playing the stimulus (line 113). The head or the ear? Turning the head and orienting an ear are two profoundly different responses, not necessarily reflecting the same reaction to the stimulus or the same cognitive elaboration of the auditory information. However, they are considered the same and merged when analyzing the results, but how was it possible to exclude that ear turning was instead elicited by any other random event?
  • Was the performance video recorded (for instance with a camera placed above the cage) in order to allow a precise assessment of both the position reached by the panda and its response to the stimuli?
  • How was exactly the turning scored? Was there a midline traced virtually on the head of the panda and some key points (nose?) taken as repere point to measure the head turn? How about the ear orientation? And how much turning was considered to include the response in the analysis?
  • Was the performance scored blindly? And by independent raters? How authors can be confident of what was measured?
  • How was the latency measured accurately, especially if there were no video recordings of the trials? 

Stimuli

  • The stimuli are unbalanced: there are 6 conspecifics calls, but only 4 heterospecific sounds; moreover, while 3 of the conspecific calls are positive and 3 are negative, all heterospecific sounds are negative. Moreover, pandas were presented with “twenty-four conspecific vocalizations and four non-conspecific sounds, i.e., twelve positive acoustic stimuli and sixteen negative stimuli” for a total of 28 stimuli for each panda. The two levels are mixed and this does allow neither a direct comparison nor a conclusion on the results if data are analyzed in the present form (see more on this on results and discussion section).
  • The repetition of each excerpt of the vocalization with a 5 s ISI (and for 5 min at maximum) could have produced ineffective stimuli because the final stimulus could have appeared too much continuous and repetitive. I know that, for instance in chicks, continuous sounds are less effective in eliciting a response behavior, independently of the relevance of the sound (De Tommaso et al., 2019). The authors should consider this and add how exactly they know that the stimuli they presented were effective in eliciting pandas’ consistent responses.

Statistics

  • Considering p<.1marginally significant is a bad practice, please provide a reference of the statistical manual where such a value is considered marginally significant. Moreover, statistical significance is a Boolean concept (i.e., 0-1), so it does not make any sense to state that something is marginally significant. The possibility authors have to discuss some values close to p<.05 is that of discussing values that approach significance.

Results

  • It remains unclear to me what is represented in figure 3. Indeed, if the values represent the average, then they cannot exceed 28, which is the number of the stimuli (and hence of the recorded responses). If the values represent the total number of responses for all pandas, then they should be around 224, but this seems not the case. They are also not percentages since the total always exceeds 100. I am definitely puzzled. Please clarify.
  • There should be an error reporting the results for the latency because the response to negative stimuli is commented identically twice, although the adjective used is different (but with the same meaning – shorter and faster – and the statistics are different (lines 162-166). Please correct and clarify.
  • I would not say that latency is faster - response times are.
  • In any of the previous cases, it is evident that, especially for positive, negative, and non-conspecific stimuli, several data points are missing. The authors do not report how many times the animals did not respond. The authors also miss discussing how they coped with missing values. An index should have probably been preferable, as it is also a common practice in all works that deal with side biases, both to score the head-orienting response of each panda to the different stimuli (L + R / L + R + Null) and lateral asymmetries in the direction of head-turning responses (L – R / L + R).
  • The categories of stimuli (with positive and negative valence and from conspecifics and non-conspecifics) cannot be directly compared as done by the authors, because non-conspecific sounds are also always negative, whereas conspecific sounds are mixed…an index should probably be preferred also for this measure.
  • In figure 4, it is not entirely clear which contrasts are shown as significant: the comparison between left positive and left negative is not significant but the asterisk suggests so.
  • Latencies are long. A latency of 4 to 7 seconds, when not 15, after the stimulus onset is a huge amount of time to orient toward a stimulus, especially if it represents a potential threat. How do the authors justify this performance? How can the authors be sure that the pandas turned after so long for the acoustic stimulus and not, instead, for some other random reason?
  • There is at least one study in the literature showing that hemispheric preference to attend to a particular class of stimuli may change depending on the exposure time (Miklosi et al., 1996). Prolonged exposure produces a shift in ear preference. How did authors cope with the fact that some animals responded earlier than others and most of them with such long latencies and hence different exposure times to the stimuli?

Discussion

  • Lines 199-201: in the authors’ words, the pandas turned leftward, but this was a trend, rather a non-significant result. What??? The pandas do not show the leftward bias for the negative stimuli, that’s all.
  • Lines 219-220: please substantiate the claim with an adequate reference. To my knowledge, this is a speculation, also rather questionable.
  • Lines 220-223: this is a further (questionable) speculation that needs at least some general reference. Indeed, while results on the lateral bias for auditory stimuli in dogs and horses could be in the direction of what authors are proposing here, there are other results on other un-domesticated species showing side biases for auditory stimuli that do not support this claim, but the authors (who know them because they cite the papers in other parts of the manuscript) do not discuss them in this section.
  • Speaking of cerebral processing and neural times (line 225) is something that has nothing to do with the present study in which no electrophysiological recording or any other measure of the brain processing was performed to assess the speed of signal processing.
  • Lines 228-229: that the right hemisphere is faster in processing emotional and conspecific stimuli, cannot be stated because the stimuli were unbalanced and hence emotional positive and conspecific sounds overlapped.
  • Lines 240-241: authors refer, for the first time in the very last line, to features that were not investigated and not even mentioned before. Specifically, familiarity: has this to do with the fact that the vocalizations presented to the pandas belonged to familiar animals sharing the facilities within the zoo? Concerning instead social and individual functions of the acoustic stimuli: what is the individual function of a sound? How is relevant for the presented data?

There are several problems with English. I urge the authors to refer to a proof editing service.

Please report all Linnaean names of the species in Italics.

DE TOMMASO M, KAPLAN G, CHIANDETTI C, VALLORTIGARA G (2019). Naïve 3-day-old domestic chicks (Gallus gallus) are attracted to discrete acoustic patterns characterizing natural vocalizations. J Comp Psychol, 133(1) 118-131

MIKLOSI A, ANDREW RJ, DHARMARETNAM M (1996). Auditory lateralisation: Shifts in ear use during attachment in the domestic chick. Laterality, 1(3) 215-224

Author Response

Response to Reviewer 4 Comments

Point 1: Introduction

Lines 48-9: if the mentioned hypotheses are relevant – and indeed they seem so because authors refer to them also in the discussion – they should be, at least briefly, described.

Response 1: Thank you for your comments, we have described briefly in line 231-232.

Point 2: Lines 51-4: it is unclear to me how the two sentences are related, please clarify.

Response 2: Thank you for your suggestion, we have corrected the two sentence in this version of line 58-60.

Point 3: Procedure

Subjects

Where were the pandas from? Details are reported for the place in which panda’s vocalizations have been recorded, but no details are available for the pandas that have been tested. If the facility where the test occurred is the same from which recordings were taken, then the playback experiment could have suffered from the familiarity of the vocalizations to the tested pandas (see also a comment about familiarity in the discussion section). They could be uninterested in the sounds if familiarity is an uncontrolled factor at play.

Response 3: Thank you for your questions, we added the tested giant panda that is from Beijing zoo, and the giant pandas from Beijing never are familiar to these vocalizations of Chengdu. We have added the information in line 86 and table 1.

Point 4: Set-up and dependent variable

I wonder why the authors chose to present the stimuli using a single speaker and waiting for the animal to align with it. The standard playback procedure used with other species implies that the animal reaches a fixed position in a standardized way and then two side-speakers broadcast at the same time the same stimulus. In this way, the sound is homogeneous and the lateral response of the animal reveals the differential elaboration carried out by the two hemispheres. The strategy adopted by Liu et al., instead, suffers from unpredictable availability of the animal (how much did the experimenter had to wait before the animal was in the correct position?), errors in alignment (how was determined that the panda was perfectly aligned to the speaker?) and bias in the subsequent response (if already in a biased position, a biased response could be the result of the procedure and not of the hemispheric differential elaboration of the stimuli).

Response 4: Thank you for your questions. The two-side speaker is one of methods to test the lateralization in dog and goat that browsing food (Siniscalchi et al. 2018; Baciadonna et al. 2019). We chose another method to test by a single speaker that conducted in sea lion, horse and chimpanzee (see Basile et al. 2009, Böye et al. 2005, Kutsukake et al. 2012), since giant pandas often sit one position to eat and do not move around. Further giant pandas are different to those animal browsing foods by mouth, and giant pandas need sit to hold the food by their paws. Thus we can settle the bamboos to a fixed position, then giant pandas could reach there to eat. We just set the speaker in the middle of its body. 

References for this answer:

Siniscalchi, M.; d'Ingeo, S.; Fornelli, S., Quaranta, A. Lateralized behavior and cardiac activity of dogs in response to human emotional vocalizations. Sci. Rep. 2018, 8, 77.

Baciadonna, L.; Nawroth, C.; Briefer, E.F.; McElligott, A.G. Perceptual lateralization of vocal stimuli in goats. Curr. Zool. 2019, 65, 67–74. https://doi.org/10.1093/cz/zoy022

Basile, M.; Boivin, S.; Boutin, A.; Blois-Heulin, C.; Hausberger, M.; Lemasson, A. Socially dependent auditory laterality in domestic horses (Equus caballus). Anim. Cogn. 2009, 12, 611–619.

Böye, M.; Güntürkün, O.; Vauclair, J. Right ear advantage for conspecific calls in  adults and subadults, but not infants, California sea lions (Zalophus californianus): Hemispheric specialization for communication? Eur. J. Neurosci. 2005, 21, 1727–1732.

Kutsukake N. Migaku Teramoto   Seijiro Homma   Yusuke Mori   Kazunari Matsudaira   Hisao Kobayashi   Takafumi Ishida   Kazuo Okanoya   Toshikazu Hasegawa   (2012). Individual variation in behavioural reactions to unfamiliar conspecific vocalisation and hormonal underpinnings in male chimpanzees. Ethology, 118(3), 269-280.

Point 5: The dependent variable was the head or the ear orientation while playing the stimulus (line 113). The head or the ear? Turning the head and orienting an ear are two profoundly different responses, not necessarily reflecting the same reaction to the stimulus or the same cognitive elaboration of the auditory information. However, they are considered the same and merged when analyzing the results, but how was it possible to exclude that ear turning was instead elicited by any other random event?

Response 5: Thank you for your questions. We reference the article on testing horse, pig and mice (Böhmer 1988, Jero et al 2001, Basile et al 2009), which the animal can react to sound by moving ears without turning the head. Although there is so far a lack of studies concerning giant panda hearing function, we also have a pre-test for the giant pandas, found the pinna of giant pandas can move when the sound stimuli were played.

References for this answer:

Basile, M.; Boivin, S.; Boutin, A.; Blois-Heulin, C.; Hausberger, M.; Lemasson, A. Socially dependent auditory laterality in domestic horses (Equus caballus). Anim. Cogn. 2009, 12, 611–619.

Jero, Coling D E , Lal A K , et al. The Use of Preyer's Reflex in Evaluation of Hearing in Mice[J]. Acta OtoLaryngologica, 2001, 121(5):585-589.

Böhmer A. The Preyer reflex--an easy estimate of hearing function in guinea pigs. Acta Otolaryngol. 1988 Nov-Dec;106(5-6):368-72. doi: 10.3109/00016488809122259. PMID: 3207004.

Point 6: Was the performance video recorded (for instance with a camera placed above the cage) in order to allow a precise assessment of both the position reached by the panda and its response to the stimuli?

Response 6: Thank you for your questions. Yes, the video recording was a precise assessment for the orientation.

Point 7: How was exactly the turning scored? Was there a midline traced virtually on the head of the panda and some key points (nose?) taken as repere point to measure the head turn? How about the ear orientation? And how much turning was considered to include the response in the analysis?

Response 7: Thank you for your questions. We identified turning head by the midline of head and ear orientation by the moving of pinna.

Point 8: Was the performance scored blindly? And by independent raters? How authors can be confident of what was measured?

Response 8: Thank you for your questions. we measured the score by a rater who do not know what meanings for this test.

Point 9: How was the latency measured accurately, especially if there were no video recordings of the trials? 

Response 9: Thank you for your questions. We design a C + + programming to play the sound stimuli, when we set to start playing, the time will be counted by computer. Once the animal has a behavioral response, we pressed the key to stop and automatically calculate the latency. All results can be output to the txt file for analysis.

Point 10: Stimuli

The stimuli are unbalanced: there are 6 conspecifics calls, but only 4 heterospecific sounds; moreover, while 3 of the conspecific calls are positive and 3 are negative, all heterospecific sounds are negative. Moreover, pandas were presented with “twenty-four conspecific vocalizations and four non-conspecific sounds, i.e., twelve positive acoustic stimuli and sixteen negative stimuli” for a total of 28 stimuli for each panda. The two levels are mixed and this does allow neither a direct comparison nor a conclusion on the results if data are analyzed in the present form (see more on this on results and discussion section).

Response 10: Thank you for your suggestion. From the comparison between conspecifics and non-conspecifics call, the stimuli are 6:4 = 3:2 that is few unbalance. From the positive and negative sounds, the stimuli are 12:16 =3:4 that is few unbalance. When we analyzed we did not mixed these valence together. We distinguished the two categories of stimuli with statistic.

Point 11: The repetition of each excerpt of the vocalization with a 5 s ISI (and for 5 min at maximum) could have produced ineffective stimuli because the final stimulus could have appeared too much continuous and repetitive. I know that, for instance in chicks, continuous sounds are less effective in eliciting a response behavior, independently of the relevance of the sound (De Tommaso et al., 2019). The authors should consider this and add how exactly they know that the stimuli they presented were effective in eliciting pandas’ consistent responses.

Response 11: Thank you for your comments. If the too much continuous and repetitive stimuli led to ineffective response, we would perform this trial after several days. We have added the consideration in line 130-131.

Point 12: Statistics

Considering p<.1marginally significant is a bad practice, please provide a reference of the statistical manual where such a value is considered marginally significant. Moreover, statistical significance is a Boolean concept (i.e., 0-1), so it does not make any sense to state that something is marginally significant. The possibility authors have to discuss some values close to p<.05 is that of discussing values that approach significance.

Response 12: We can appreciate the reviewer’s viewpoint about p values and statistical significance. It is widely taught in statistics courses the p < 0.05 indicates statistical significance, but rarely taught that a p value slightly larger than 0.05 offers suggestive evidence against the null hypothesis of no effect. Many prominent statisticians have been trying to correct this practice, as it is overly restrictive. The practice leads to more type II errors (failing to find significance) in favor of avoiding Type I errors (finding false significance), whereas the researcher should be concerned about both types of errors. Here is a short list of widely quoted works with references: Fisher, himself, who introduced hypothesis testing in 1935, establishing the convention of p < 0.05 to indicate statistical significance, later in 1956 retracted the idea as “absurdly academic”, advocating, instead, that actual values of the p values should be shared with fellow researchers (Gigerenzer et al., 2004). Burdette and Gehan (1970, p.9), Utts and Heckard (2006, p.361) later advocated that 0.05< p< 0.10 provided “suggestive evidence” of statistical significance and should be reported as such.

Altman, Gore, Garder, and Pocock (1983, section 4.3) advise, “Calling any value with p > 0.05 ‘not significant’ is not recommended, as it may obscure results that are not quite statistically significant but do suggest a real effect (see section 5.1).”, and clarify further (1983, section 5.1) with, “Some flexibility is desirable in interpreting p values. The 0.05 level is a convenient cut off point, but p values of 0.04 and 0.06, which are not greatly different, ought to lead to similar interpretations, rather than radically different ones. The designation of any result with p > 0.05 as not significant may thus mislead the reader (and the authors); hence the suggestion in section 4.3 to quote actual p values.”

Altman (1991, pp.168-169) states that “The cut-off level for statistical significance is usually taken at 0.05, but sometimes at 0.01. These cut-offs are arbitrary and have no specific importance. It is ridiculous to interpret the results of a study differently, according to whether the p value obtained was, say, 0.055 or 0.045. These p values should lead to very similar conclusions, not diametrically opposed ones. A minor change to the data can easily shift the p value by this amount or more….Quoting the actual p value allow the reader to make his or her own interpretation.”

These arguments are obviously logical and are slowly gaining acceptance. We can find the practice of allowing the reporting of “marginally significant” or “trend towards significant” statements in a lot of top journals (Karunajeewa et al, 2008; Matsuzaki et al, 2010; Sharkey 2010; Stahl et al, 2012). In these studies, significance was defined as p < 0.05, and marginal significance was defined as 0.05 ≤ p < 0.10 in statistical analysis. The same definitions had been declared in our last version. Furthermore, the actual p values were reported, so the readers can assess the evidence and decide for himself or herself, so we would like to keep the sentences unchanged.

References for this answer :

Altman DG (1991) Practical Statistics for Medical Research. New York, Chapman & Hall/CRC.

Altman DG, Gore SM, Gardner MJ, Pocock SJ (1983) Statistical guidelines for contributors to medical journals. British Medical Journal (Clinical Research Ed) 286: 1489-1493.

Burdette WJ, Gehan EA (1970) Planning and Analysis of Clinical Studies, Springfield IL: Charles C. Thomas.

Gigerenzer G, Krauss S, Vitouch O (2004) In D. Kaplan (ed.) The Sage Handbook of Quantitative Methodology for the Social Sciences, Thousand Oakes, CA, Sage, pp. 391-408.

Karunajeewa HA, Mueller I and Senn M, et al. (2008) A trial of combination antimalarial therapies in children from Papua New Guinea. New England Journal of Medicine 359: 2545-57.

Matsuzaki T, Sasaki K, Tanizaki Y, Hata J, Fujimi K, et al. (2010) Insulin resistance is associated with the pathology of Alzheimer disease The Hisayama Study. Neurology 75: 764-770.

Sharkey P (2010) The acute effect of local homicides on children's cognitive performance. Proceedings of the National Academy of Sciences 107: 11733-11738.

Stahl EA, Wegmann D, Trynka G, Gutierrez-Achury J, Do R, et al. (2012) Bayesian inference analyses of the polygenic architecture of rheumatoid arthritis. Nature Genetics 44: 483-491.

Utts JM and Heckard RF (2006) Statistical Ideas and Methods, Belmont, CA: Thomson Brooks/Cole.

Point 13: Results

It remains unclear to me what is represented in figure 3. Indeed, if the values represent the average, then they cannot exceed 28, which is the number of the stimuli (and hence of the recorded responses). If the values represent the total number of responses for all pandas, then they should be around 224, but this seems not the case. They are also not percentages since the total always exceeds 100. I am definitely puzzled. Please clarify.

Response 13: Thank you for your questions, we are very sorry that we filled the wrong vertical ordinate to “Mean number of responses”. Actually it is “total number of responses” for all pandas. We have corrected the Figure 3 and added the explanation of legend in this version.

Point 14: There should be an error reporting the results for the latency because the response to negative stimuli is commented identically twice, although the adjective used is different (but with the same meaning – shorter and faster – and the statistics are different (lines 162-166). Please correct and clarify.

Response 14: Thank you for your questions. The first report is the result of all orientation that latency of negative sounds was shorter than that of positive sounds. The second report is the result of turning left that latency of negative sounds was shorter than that of positive sounds. Therefore, it is not twice for reporting. Here is “The latency of the response to negative stimuli was significantly shorter than that of the response to positive stimuli (Wilcoxon signed ranks test: Z=−2.277, P=0.023) (Fig. 4), especially for turning to the left side, and the latency of the response to negative stimuli was significantly shorter than that of the response to positive stimuli (Wilcoxon signed ranks test: Z=−2.841, P=0.004) (Fig. 4).”

Point 15: I would not say that latency is faster - response times are.

Response 15: Thank you for your comments, we have changed all latency adjective words to faster, the slower to longer.

Point 16: In any of the previous cases, it is evident that, especially for positive, negative, and non-conspecific stimuli, several data points are missing. The authors do not report how many times the animals did not respond. The authors also miss discussing how they coped with missing values. An index should have probably been preferable, as it is also a common practice in all works that deal with side biases, both to score the head-orienting response of each panda to the different stimuli (L + R / L + R + Null) and lateral asymmetries in the direction of head-turning responses (L – R / L + R).

Response 16: Thank you for your suggestions. In this revised version, we calculate (L + R / L + R + Null) and (L – R / L + R) for each panda and each type of stimuli (positive, negative, conspecific and non-conspecific sounds and all sounds), and then we use One-sample Wilcoxon Signed Rank Test to compare these values with 1 and 0, respectively. The statistical results show that, for each type of stimuli, the value of (L + R / L + R + Null) is significantly or marginally significantly greater than 1, however, there is no significant difference between the value of (L – R / L + R) and zero. Small sample size in the present study may be the reason why there are significant differences for response bias according to number of response but not laterality index.

Point 17: The categories of stimuli (with positive and negative valence and from conspecifics and non-conspecifics) cannot be directly compared as done by the authors, because non-conspecific sounds are also always negative, whereas conspecific sounds are mixed…an index should probably be preferred also for this measure.

Response 17: Thank you for your suggestions. Although non-conspecific sounds are all negative, conspecific sounds include the negative and positive, so it is different from the comparison between negative stimuli and positive stimuli. Just like the studies of sea lion were divided into four categories regarding their origin (conspecific, Csp; or non-specific, Nsp) and their degree of familiarity (familiar, F or unfamiliar, U) on sea lion (Boye et al. 2005).

Böye, M.; Güntürkün, O.; Vauclair, J. Right ear advantage for conspecific calls in  adults and subadults, but not infants, California sea lions (Zalophus californianus): Hemispheric specialization for communication? Eur. J. Neurosci. 2005, 21, 1727–1732.

Point 18: In figure 4, it is not entirely clear which contrasts are shown as significant: the comparison between left positive and left negative is not significant but the asterisk suggests so.

Response 18: Thank you for your carefully suggestion, Although the left positive value and left negative value is 3.9412 vs 3.9077, which is too close, but the comparison between left positive and left negative is significant in statistic. In addition, we revise this figure 4 to be clear in this version.

Point 19: Latencies are long. A latency of 4 to 7 seconds, when not 15, after the stimulus onset is a huge amount of time to orient toward a stimulus, especially if it represents a potential threat. How do the authors justify this performance? How can the authors be sure that the pandas turned after so long for the acoustic stimulus and not, instead, for some other random reason?

Response 19: Thank you for your questions. After the response of a negative stimulus, we usually stop this test for 5 mins so that giant pandas have a break and recover from that of stimuli.  During the trials, we kept the environments and situation are very quite and homogenous, which make sure that there is no disturbance from the other random reasons. Therefore, the latency of near 15s can be sure that it comes from the response of giant pandas.

Point 20: There is at least one study in the literature showing that hemispheric preference to attend to a particular class of stimuli may change depending on the exposure time (Miklosi et al., 1996). Prolonged exposure produces a shift in ear preference. How did authors cope with the fact that some animals responded earlier than others and most of them with such long latencies and hence different exposure times to the stimuli?

Response 20: Thank you for your questions. We checked the data that the mean latency of young is 6.4961s, while the old is 9.4557s. Although the sample size is limited, there is a significant by U test as follows. Therefore, we think the longer latencies come from the difference of age, and it is not from shifting the ear preference.

Point 21: Discussion

Lines 199-201: in the authors’ words, the pandas turned leftward, but this was a trend, rather a non-significant result. What??? The pandas do not show the leftward bias for the negative stimuli, that’s all.

Response 21: Thank you for your question, yes, the pandas only show the rightward bias for the positive stimuli, and for the negative stimuli there is no significant result, but the latency of negative stimuli was significant. We have mentioned in line 171-176 of this version.

Point 22: Lines 219-220: please substantiate the claim with an adequate reference. To my knowledge, this is a speculation, also rather questionable.

Response 22: Thank you for your comment. We have deleted this speculative sentence in this version.

Point 23: Lines 220-223: this is a further (questionable) speculation that needs at least some general reference. Indeed, while results on the lateral bias for auditory stimuli in dogs and horses could be in the direction of what authors are proposing here, there are other results on other un-domesticated species showing side biases for auditory stimuli that do not support this claim, but the authors (who know them because they cite the papers in other parts of the manuscript) do not discuss them in this section.

Speaking of cerebral processing and neural times (line 225) is something that has nothing to do with the present study in which no electrophysiological recording or any other measure of the brain processing was performed to assess the speed of signal processing.

Response 23: Thank you for your comments. We revised this section and deleted the parts of speculation, then we cited three family of carnivore to discuss in line224-231.

Point 24: Lines 228-229: that the right hemisphere is faster in processing emotional and conspecific stimuli, cannot be stated because the stimuli were unbalanced and hence emotional positive and conspecific sounds overlapped.

Response 24: Thank you for your questions, this problem has been explained in parts of stimuli. Although conspecific sounds and positive of negative have an overlapped part, we analyzed this valence to two different groups: one is emotion, another is conspecifics and non-conspecifics, which there is no contradiction and repeat.

Point 25: Lines 240-241: authors refer, for the first time in the very last line, to features that were not investigated and not even mentioned before. Specifically, familiarity: has this to do with the fact that the vocalizations presented to the pandas belonged to familiar animals sharing the facilities within the zoo? Concerning instead social and individual functions of the acoustic stimuli: what is the individual function of a sound? How is relevant for the presented data?

Response 25: Thank you for your questions. We have mentioned this vocalization is from Chengdu zoo where is a thousands miles to Beijing, and the pandas never share the facilities, which is not familiar giant pandas for tested subject animals. We have added this information in line 94-95.We considered that the valence of familiarity did not include in our study, though the familiarity word appeared the in the last line.

Point 26: There are several problems with English. I urge the authors to refer to a proof editing service.

Response 26: Thank you, we have checked English again.

Point 27: Please report all Linnaean names of the species in Italics.

Response 27: Thank you for your suggestions, we are very sorry for the mistakes. We have changed all Latin names to italic.

END

Round 2

Reviewer 1 Report

The authors have integrated all comments in a meaningful way.

Author Response

Dear Reviewer,

Thank you so much for your positive suggestions in this version. We invited a native English speaker to correct English. We have clarified some sentences as questioned and touched up English. Hopefully this version can be satisfied as the requirement of you and the journal.

Best Regards

He Liu  Yezhong Tang  Yanxia Ni  Guangzhan Fang

Reviewer 4 Report

In this re-review I want to focus on the answers and arguments of the authors in their rebuttal letter. 

Why have the Authors not included the procedural details in the revised version of the manuscript? It is not that they have to convince me, rather they have to convince a potential reader whose skepticism I anticipated in my first review. Why is there no mention of the presence of a camera and the subsequent offline scoring of the response, how the scoring was performed, how they did proceed to keep the experimenter blind to the conditions/scope of the experiment, etc.?

Inter-raters reliability is completely missing and I do not understand the reason why, having the video recordings, Authors did not add evidence that the scoring was consistent across independent judges. The agreement on what is observed is fundamental in the domain of ethological observations but here the measurements seem highly arbitrary and discretionary. Indeed, the Authors also wrote “We identified turning head by the midline of head and ear orientation by the moving of pinna.” but this is not a specification of the procedure followed. Nobody by reading this procedure would be able to reproduce it. Did the Authors use a software to trace reference points? Did they manually superimpose a transparent sheet over the monitor to trace a midline? How was considered an inclination in the z-axis? I could continue...

As for the dependent variable, the Authors write: “the animal can react to sound by moving ears without turning the head.” Then, as per admission by the Authors themselves, the two responses are independent, in line with what I suggested.

Considering the latencies, the Authors responded only to the second part of my criticism. Indeed, the fact that 15 seconds is a huge amount of time to respond to an acoustic stimulus, especially in the case of sounds with negative valence, is unquestionable from an ecological stance.

Moreover, an imbalance is an imbalance - in the choice and presentation of stimuli, no matter how little it is. The work by Boyle et al is different and, anyway, it cannot be used as a justification for the current work, if there is a mistake.

As for the statistics: a possible alternative response to that provided by the Authors was to consider adding further details on the analysis performed. The reviewer still sees value to get more informative results by adding for instance a power analysis that provides the reader with a real significance of the p-value. Such an approach would add quality and value to the analysis. Also, a Bayesan approach should have been preferred if the Authors feel so uncomfortable with Boolean p considering that this approach helps determine which of the two hypotheses has more support given the data.

Given the concerns that still remain after the revision and the somewhat surprising resistance of the authors to improve the study and/or the manuscript (more procedural details, more precision and quality in the definitions and the scoring, more raters, more analysis, more data points), I am sorry to feel unable to recommend the paper for publication in ANIMALS.

Author Response

Dear reviewer,

Thank you for your efforts on our manuscript. We really appreciate for your patient work on our manuscript. After carefully understanding your final comments, we realized that the details of measure are very fundamental for this study and the readers. So we carefully corrected the description of methods and tried to mention the procedure of judgement in detail. However, there are some limitations existed that the small sample size and the unbalanced stimuli. All the changes have been replaced as you suggested in the manuscript. We tried our best to improve the paper as you noted in the PDF version.

   We are very pleased that our paper has been given by a chance for further suggestions and revisions. We are very grateful to you for your kindly considering our MS.

Sincerely yours

Comments and Suggestions for Authors

In this re-review I want to focus on the answers and arguments of the authors in their rebuttal letter.

Point 1:

(1) Why have the Authors not included the procedural details in the revised version of the manuscript? It is not that they have to convince me, rather they have to convince a potential reader whose skepticism I anticipated in my first review. (2) Why is there no mention of the presence of a camera and the subsequent offline scoring of the response, (3) how the scoring was performed, (4) how they did proceed to keep the experimenter blind to the conditions/scope of the experiment, etc.?

Response 1:

  • We really feel sorry for no highlighted version since we found the number of line is missing so we accepted these revisions. We have updated procedural details in this revised version.
  • We are also sorry that we did not mention clearly about the camera at the first revision. We preformed the scoring of response on the spot and did not apply the video for the offline scoring of response. Following your advices, we wanted to make a confirm test by video, but the part of data was missing.
  • After a scoring of response by the experimenter, we further conducted a scoring confirm on the spot and then record. We already added this procedure in line 173-188.

(4) We chose an experimenter who was blind to the anticipative response for a given acoustic   stimulus and the meaning of the data being collected. Before scoring of animals’ response we have a training to tell the situation of orientation for the experimenter and make a reliability test with the other researchers. The reliability test is that we play some sounds and then let the experimenter and the researchers separately judge, and evaluate the similarity rate to 100%.

Point 2: (1) Inter-raters reliability is completely missing and I do not understand the reason why, having the video recordings, Authors did not add evidence that the scoring was consistent across independent judges. The agreement on what is observed is fundamental in the domain of ethological observations but here the measurements seem highly arbitrary and discretionary. Indeed, the Authors also wrote “We identified turning head by the midline of head and ear orientation by the moving of pinna.” but this is not a specification of the procedure followed. Nobody by reading this procedure would be able to reproduce it. (2)Did the Authors use a software to trace reference points? (3)Did they manually superimpose a transparent sheet over the monitor to trace a midline? (4)How was considered an inclination in the z-axis? I could continue...

Response 2:

  • We are sorry for missing description of the inter-raters reliability, which is a classical measure and basic agreement for observing behavior and scoring response. We have added these in line 184-185. We make a final confirm on the spot by the researchers in case of inaccurate video recording, so we did not mention the video.
  • We did not trace point using software.
  • We did not use a transparent sheet since we have a middle object to reference. We tried to adjust the experimenter keeping on a straight line with the subject panda according to the object of reference. We have added further details in line185-186.
  • We did not use the z-axis to scoring of response.

Point 3: As for the dependent variable, the Authors write: “the animal can react to sound by moving ears without turning the head.” Then, as per admission by the Authors themselves, the two responses are independent, in line with what I suggested.

Response 3: Thank you for your attention. We chose an indoor environment to perform this test, where there is no disturbation from noise and the other external interference. Considering the responses from acoustic stimuli, we think that the two responses could be treated as the same response of giant panda.

Point 4: Considering the latencies, the Authors responded only to the second part of my criticism. Indeed, the fact that 15 seconds is a huge amount of time to respond to an acoustic stimulus, especially in the case of sounds with negative valence, is unquestionable from an ecological stance.

Response 4: Thank you for your attention. As your comments, 15 seconds is a long time to response negative sounds, but our results shows that this time only response to a positive valence. Potentially, a huge amount of time in response to acoustic stimuli may be associated with very limited negative experience in the individuals kept in a zoo. It is possible that the captive giant pandas did not get a chance to engage in negative social interactions or experience of some negative situations.

Point 5: Moreover, an imbalance is an imbalance - in the choice and presentation of stimuli, no matter how little it is. The work by Boyle et al is different and, anyway, it cannot be used as a justification for the current work, if there is a mistake.

Response 5: Following your advice, we accepted this imbalance of stimuli, but we do not think it is a mistake that can be affected the results. However, we will consider this balance in future study. We also added a limitation at the end of discussion in line 382-388.

Point 6: As for the statistics: a possible alternative response to that provided by the Authors was to consider adding further details on the analysis performed. The reviewer still sees value to get more informative results by adding for instance a power analysis that provides the reader with a real significance of the p-value. Such an approach would add quality and value to the analysis. Also, a Bayesan approach should have been preferred if the Authors feel so uncomfortable with Boolean p considering that this approach helps determine which of the two hypotheses has more support given the data.

Response 6:

  • Thank you for your suggestions. In the revised version, we calculate effect size for single-sample chi-square test and Wilcoxon signed ranks test, respectively. The following sentence has been added to the section ‘Statistical analyses’, “Effect size was determined with Cohen's w and r for single-sample chi-square test and Wilcoxon signed ranks test respectively, and that the value 0.10 as a small, 0.30 as a medium and 0.50 as a large effect size, respectively.” The values of effect size have been added to the main text in line 217-222.
  • Because the data failed to meet the statistical assumptions such as multivariate normal distribution and homogeneity of variance, we do not analyze the data using Bayesian approach.  

Point 7: Given the concerns that still remain after the revision and the somewhat surprising resistance of the authors to improve the study and/or the manuscript (more procedural details, more precision and quality in the definitions and the scoring, more raters, more analysis, more data points), I am sorry to feel unable to recommend the paper for publication in ANIMALS.

Response 7: We are very sorry for our limited interpretation and language understanding in haste. We would like to thank you for your careful review and insightful comments, which improved to be a preferably scientific level. We have added clarifications to the report in each section following your valuable advice, and a lot of changes have been taken place. We are very grateful to you for your kindly considering our corrected manuscript. We really cherish the opportunity if the MS can be published at ANIMALS.

Round 3

Reviewer 4 Report

-

Author Response

Dear Reviewer,

Thank you for your comments. We really appreciate your professional efforts on our manuscript. This version of MS was checked the English again and corrected the spelling and grammar carefully and made correction as follow: in line 11-18,line 137-157;200-201;348-350.

In addition, some minor syntax errors have also been corrected. We hope that this version of manuscript can meet with the approval.

Happy Lunar New Year of the Ox

This manuscript is a resubmission of an earlier submission. The following is a list of the peer review reports and author responses from that submission.